# Epidemiological and virological factors determining dengue transmission in Sri Lanka during the COVID-19 pandemic

Dinuka Ariyaratne[1], Laksiri Gomes[1], Tibutius T. P. Jayadas[1], Heshan Kuruppu[1], Lahiru Kodituwakku[2], Chandima Jeewandara[1], Nimalka Pannila Hetti[2], Anoja Dheerasinghe[2], Sudath Samaraweera[2], Graham S. Ogg[3], Gathsaurie Neelika Malavige[1,3]*

1 Department of Immunology and Molecular Medicine, Allergy Immunology and Cell Biology Unit, University of Sri Jayewardenepura, Nugegoda, Sri Lanka, 2 National Dengue Control Unit, Ministry of Health, Colombo, Sri Lanka, 3 Human Immunology Unit, MRC Weatherall Institute of Molecular Medicine, University of Oxford, Oxford, United Kingdom

* gathsaurie.malavige@ndm.ox.ac.uk

**Data Availability Statement:** All data is available in the manuscript and the figures.

## Abstract

With the onset of the COVID-19 pandemic in early 2020 there was a drastic reduction in the number of dengue cases in Sri Lanka, with an increase towards the end of 2021. We sought to study the contribution of virological factors, human mobility, school closure and mosquito factors in affecting these changes in dengue transmission in Sri Lanka during this time. To understand the reasons for the differences in the dengue case numbers in 2020 to 2021 compared to previous years, we determined the association between the case numbers in Colombo (which has continuously reported the highest number of cases) with school closures, stringency index, changes in dengue virus (DENV) serotypes and vector densities. There was a 79.4% drop in dengue cases from 2019 to 2020 in Colombo. A significant negative correlation was seen with the number of cases and school closures (Spearman's r = -0.4732, p <0.0001) and a negative correlation, which was not significant, between the stringency index and case numbers (Spearman's r = -0.3755 p = 0.0587). There was no change in the circulating DENV serotypes with DENV2 remaining the most prevalent serotype by early 2022 (65%), similar to the frequencies observed by end of 2019. The Aedes aegypti premise and container indices showed positive but insignificant correlations with dengue case numbers (Spearman r = 0.8827, p = 0.93). Lockdown measures, especially school closures seemed to have had a significant impact on the number of dengue cases, while the vector indices had a limited effect.

## Introduction

Dengue is the most common mosquito borne viral infection in Sri Lanka and in many tropical and sub-tropical countries. There has been over a 20-fold rise in the number of dengue cases during the last 20 years in Sri Lanka, with the number of cases and deaths rising

**Funding:** We are grateful to the Accelerating Higher Education Expansion and Development (AHEAD) Operation of the Ministry of Higher Education funded by the World Bank (GNM), Centre for Dengue Research (GNM) and the UK Medical Research Council (GSO). The funders had no role in the study design and writing the manuscript.

**Competing interests:** The authors have declared that no competing interests exist.

disproportionately among adults, compared to children [1]. Similar rises in the number of cases and deaths have been seen in many Asian and Latin American countries, with many experiencing yearly epidemics [2]. Changes in the dengue virus (DENV) serotypes, population immunity and other socio-ecological factors such as changes in land use, urban poverty and human movement have shown to play an important role in the incidence of dengue [2].

With the emergence of the COVID-19 pandemic, many countries reported significantly fewer dengue cases in both year 2020 and 2021. Many countries in South East Asia and Latin America, excluding Singapore, Brazil and Peru had reported a 44.1% reduction in the reported number of dengue cases by the end of 2020, compared to previous years [3]. This reduction was shown to be largely due to a school closures and restriction in human movement to non-residential areas, rather than due to under reporting [3]. Although, Sri Lanka, India, Nepal, Indonesia and certain other South Asian and South East Asian countries were excluded from this analysis, India experienced a 84% reduction in the number of cases in 2020 [4], while Sri Lanka experienced a 74% reduction from March 2020 to April 2021 [5]. While one study attributed the decline in the number of cases of dengue in Sri Lanka due to reduction in human mobility [6], another study showed that there was a 88.6% reduction in the Aedes aegypti larvae collected from ovitraps in a Northern district in Sri Lanka [5]. Therefore, many factors such as reduced human mobility, reduction in vector density and changes in DENV serotypes could have resulted in a reduction in dengue cases experienced by different countries.

In the studies which explored the reasons for decline of dengue in South East Asia and Latin America the role of the changes in the DENV serotypes in the change dengue cases was not investigated. Therefore, it would be important to study the relationship between DENV serotypes and dengue cases in a period of limited human mobility and possibly changed dengue transmission to better understand how DENV serotypes evolve in a DENV endemic country. Therefore, in order to fully understand the factors that lead to reduction in dengue cases in Sri Lanka, we investigated the relationship between dengue cases reported in the Colombo district, which has reported the highest number of cases in Sri Lanka so far, with school closures, stringency index, changes in DENV serotypes and vector densities.

## Materials and methods

### Collection of samples from patients with acute dengue infection

Real time PCR for the four DENVs were carried out in 297 blood sample of adult patients with clinically suspected to have an acute dengue infection, following informed consent. The samples were collected from the Army Hospital Colombo, Colombo South Teaching Hospital, National Institute for Infectious Diseases, Sri Jayewardenepura General Hospital, in the Western Province of Sri Lanka, from June 2021 to January 2022 as dengue disease surveillance activities carried out by the National Dengue Control Unit Sri Lanka along with the Centre for Dengue Research, University of Sri Jayewardenepura. Details of clinical disease severity was obtained when the patient was discharged from the hospital and patients were classified as having dengue fever (DF) or dengue haemorrhagic fever (DHF), based on the WHO clinical disease classification guidelines [7]. Accordingly, 220 had DF and 39 had DHF, and in 38 clinical disease severity was not recorded.

Ethics approval for the study was obtained from the Ethics Review Committee, University of Sri Jayewardenepura and the administrative clearance was obtained from the Ministry of Health, Sri Lanka. All individuals gave informed written consent.

### Numbers of dengue infections and SARS-CoV-2 infections reported each month from the Western Province in Sri Lanka

The National Dengue Control Unit (NDCU) carries out active dengue surveillance by obtaining patient data from all clinically suspected patients with dengue infection admitted to state hospitals and private hospitals, using a platform named the DenSys [8]. The monthly case numbers of all clinically suspected patients with an acute dengue infection were obtained from this national dengue surveillance system (DenSys) [8]. The system of dengue infection surveillance of sentinel hospitals (DenSys) reports clinically suspected dengue infection, the confirmed case numbers were obtained from epidemiological surveillance reports [9]. The number of individuals infected with the SARS-CoV-2 virus since March 2020 was obtained from the COVID-19 epidemiological summary, published by the Epidemiology Unit, Ministry of Health, Sri Lanka [10].

### The stringency index of the Western Province since onset of the COVID-19 pandemic

Due to the COVID-19 pandemic, the Sri Lankan government imposed several lockdowns and variable degrees of movement restrictions, closure of schools, commercial establishments and stay at home orders. Data regarding the stringency index since March 2020 was taken from Our World in Data [11]. The stringency index is a measure of how strict a country's policies on human mobility restriction are and it takes into account nine indices for this calculation. Namely, workplace closures, cancellation of public events, restrictions on public gatherings, closures of public transport, stay-at-home requirements, public information campaigns, restrictions on internal movements and international travel controls. In addition, as we wished to understand the possible relationship between school closure and number of dengue cases, we obtained the data regarding the months of full school closures and partial closure of Sri Lankan schools from January 2021 to December 2021 [12]. While full closure indicates all schools were closed, partial school closure indicated that either only some areas had schools open, or only children of certain grades attended school.

### Mosquito indices

In order to understand the relationship of dengue case numbers with vector densities, data regarding the mosquito indices and contained indices was obtained from the NDCU, which carried out monthly entomological surveillance in different geographical areas of the Western Province.

The mosquito vector indices were calculated as follows:

The Premise Index (PI) for Aedes albopictus and Aedes aegypti were calculated as follows.

For Aedes aegypti PI:

PI = (Aedes aegypti larvae positive premises/Number of premises inspected) *100

For Aedes albopictus PI:

PI = (Aedes albopictus larvae positive premises / Number of premises inspected) *100

The container index (CI) for Aedes albopictus and Aedes aegypti were calculated as follows:

For Aedes aegypti:

CI = (Aedes aegypti larvae positive containers /Number of wet containers inspected) * 100

For Aedes albopictus

CI = (Aedes albopictus larvae positive containers/ Number of wet containers inspected)* 100

### Real time qPCR for detection of the DENV and serotypes

Viral RNA in serum was extracted using QIAamp Viral RNA Mini Kit (Qiagen, USA, Cat: 52906). Quantitative real-time PCR (RT-qPCR) was performed as previously described using the Centers for Disease Control and Prevention (CDC) real time PCR assay for detection of the DENV [13] Oligonucleotide primers and a dual labeled probe for DENV 1–4 serotypes were used (Life technologies, USA) based on published sequence [13] with slight modifications. Real-time PCR was performed using TaqMan Multiplex Master Mix (Applied Biosystems, USA, Cat: 4461881). The reactions consisted of 25 µl volumes and contained the following reagents, 1×TaqMan multiplex master mix (containing Mustag Purple dye), 900 nM of each primer, 250 nM of each probe, 7 µl of RNA and PCR grade water (Applied Biosystems, USA, Cat: AM9935). After the primers and probes were validated, a multiplex method was optimized to quantify the four serotypes in a single reaction. A total of 27 infections were DEN1, 122 were DEN2 and 28 were DEN3. No DEN4 infections were recorded and 91 were RT-PCR negative.

### Statistical analysis

Statistical analysis was carried out using the GraphPad Prism (Version 9.3.1) and non-parametric statistical tests were used. The association between the number of dengue cases with the stringency index, container index (CI) premise index (PI) and school closures were computed using the Spearman rank order correlation coefficient. All tests were two tailed.

## Results

### Changes in the number of dengue cases in the 2020 and 2021 years

Sri Lanka has been experiencing epidemics of dengue infection since 1989, with the number of cases gradually increasing over time [1]. The number of annual dengue cases have remained over 35,000 for the past 10 years and since the 2017 outbreak due to the DENV2 serotype, annual cases have exceeded 50,000 [9], with adults predominantly being infected compared to children [1]. However, during the year 2020, when Sri Lanka experienced a very strict lockdown, the total number of dengue cases in Sri Lanka was only 34151 for 2020 and 4257 for Colombo (Fig 1A and 1B). Although most dengue cases are usually seen in the Colombo district (19.8% in 2018 and 19.7% in 2019) in 2020, the highest number of cases (19.2%) was seen in the Eastern province. In addition, the dengue outbreaks usually coincide with the monsoon seasons in Sri Lanka, and this increase in the cases with the monsoons was not seen during 2020 and 2021 (Fig 1B). The number of cases recorded in Colombo 2020 (4247) is 79.4% less than in 2019. This lower number of dengue cases continued throughout 2021 with case numbers rising towards the end of the year.

### Differences in circulating DENV serotypes from 2015 to 2022

Until mid-2016, DENV1 was the predominant DENV serotype, which was replaced by DENV2 during 2017 [1]. Following the massive outbreak that occurred in 2017, the predominant DENV serotype remained as DENV2 until late 2019. In 2019 a total of 114240 cases were recorded, with the number of cases gradually increasing towards the latter part of year (Fig 1A). DENV3 was seen to emerge towards the end of 2019, which coincided with the increase in the number of cases and DENV3 accounted for 28.9% of infections in the Colombo district by end of December 2019 (Fig 2A). With the first cases of COVID-19 being reported in Sri Lanka in early 2020, Sri Lanka started a strict lockdown from 20[th] March 2020. As there were very limited numbers of admissions of patients due to dengue until June 2021, serotyping of

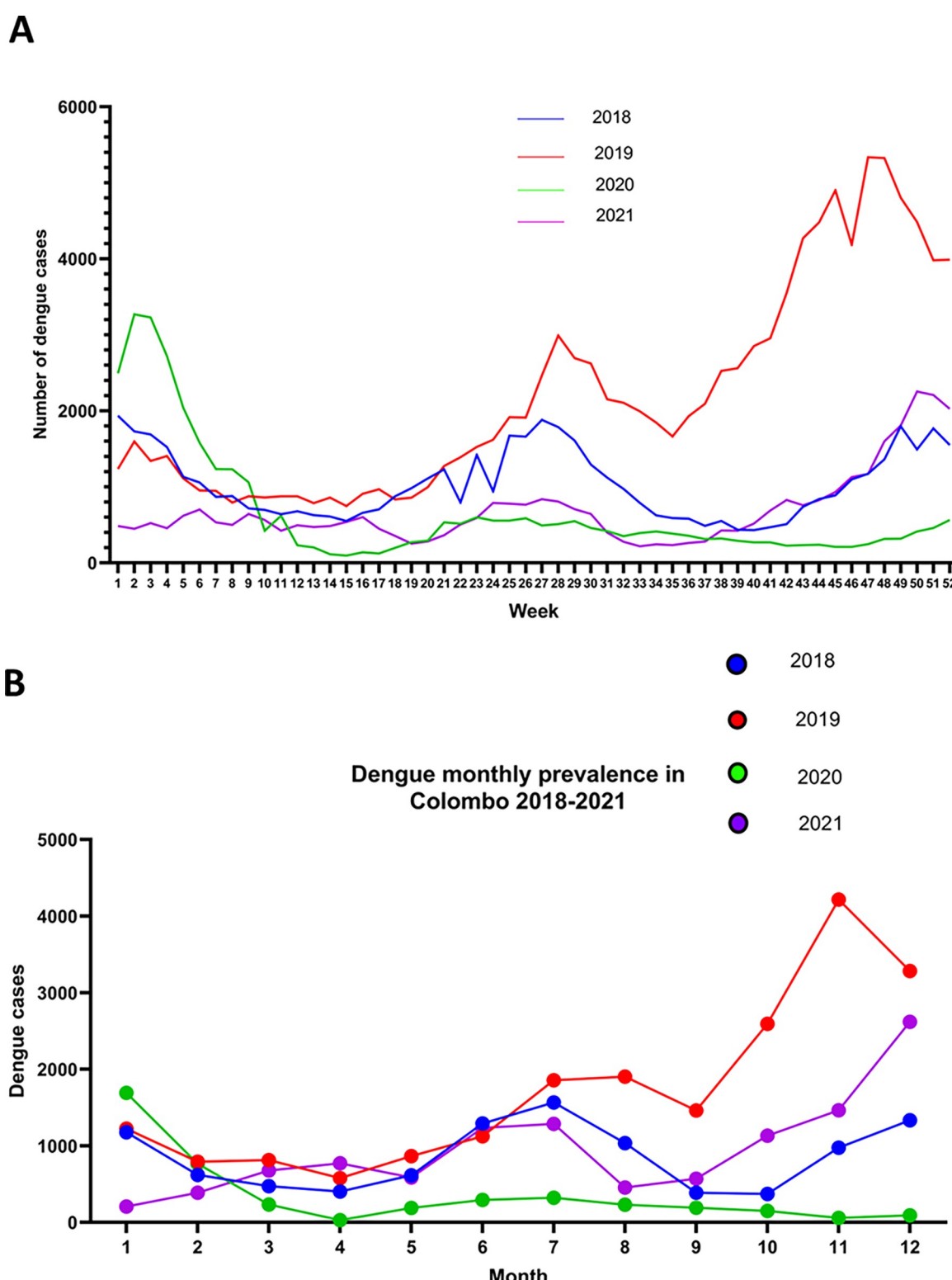

**Fig 1. The number of dengue cases reported from 2018 to 2021.** The weekly reported number of dengue cases for Sri Lanka from 2018 and 2019 (A) and the weekly number of dengue cases reported in the Colombo district (B), was obtained from the National Dengue Control Unit (NDCU) active dengue surveillance platform DenSys, which collects data from sentinel hospitals and epidemiological surveillance.

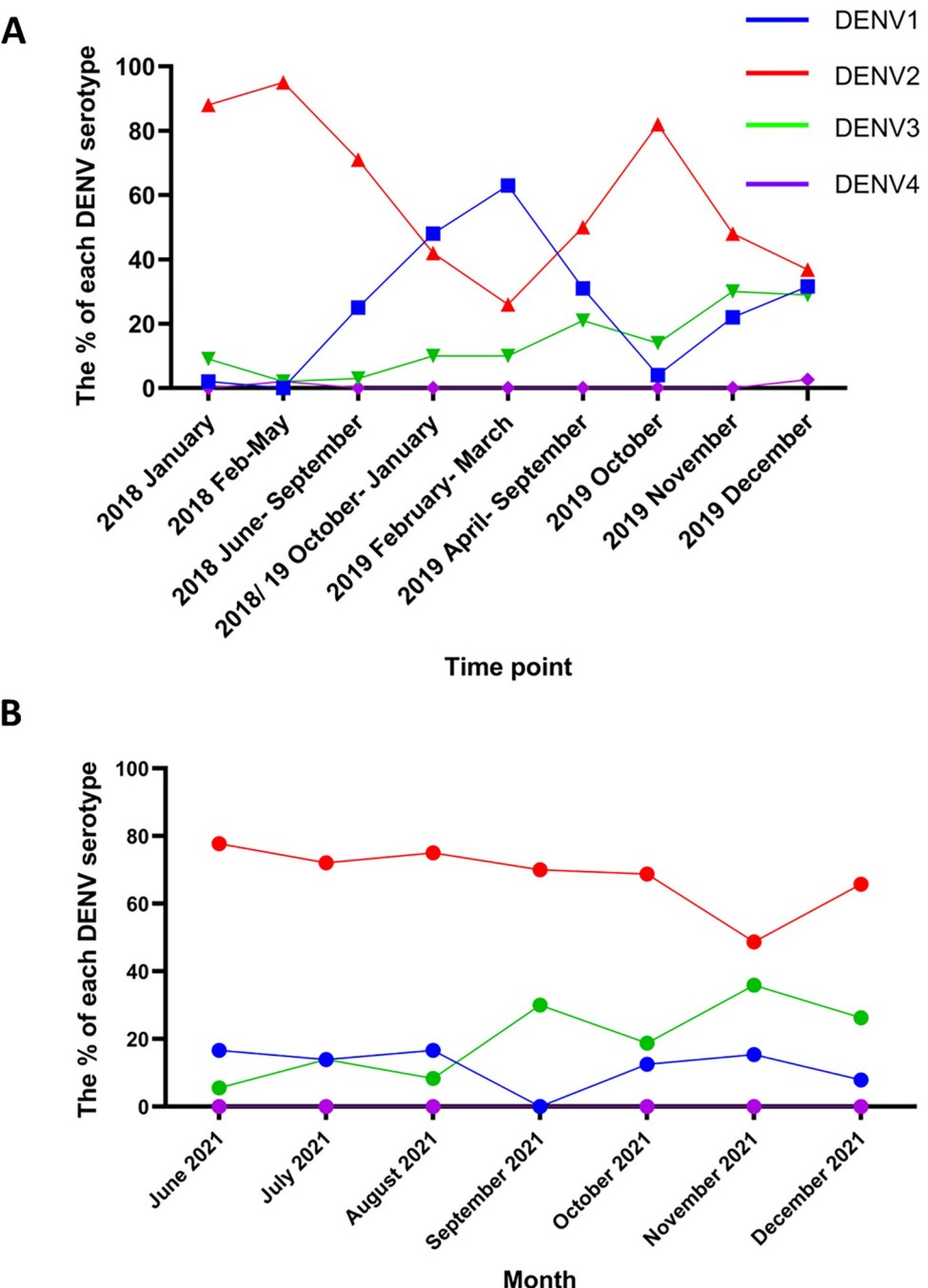

**Fig 2. Changes in the circulating dengue virus serotypes in Colombo from 2018 to 2021.** Serotyping of the DENVs was carried out on sera obtained from patients with acute dengue, in the Colombo district from 2018 to end of 2019 (A) and from June 2021 to December 2021(B). The percentage of each DENV serotype is shown.

DENVs in these patients could not be carried out from 2020 March to 2021 June. Since 2021 August, the number of dengue cases gradually rose and DENV2 was found to be the predominant circulating serotype, followed by DENV3 and DENV1 (Fig 2B). By January 2022, 65% of infections in Colombo were due to DENV2, followed by 26% of DENV3 and 7% of DENV1. Of the 220 patients with DF, 27 (12.2%) had DENV1, 105 (47.7%) had DENV2 and 24 (10.9%)

had DENV3. Out of the 39 patients who had DHF, 0 had DENV1, 17 (43.5%) had DENV2, 4 (10.2%) had DENV3 and in 18 cases (46.1%) the PCR was negative for all serotypes. In 38 patients the disease severity had not been recorded.

## Relationship between school closure, stringency index and dengue cases in 2020 and 2021

The stringency index of Sri Lanka from March 2020 to December 2021, was obtained from Our World in Data [11]. This is a measure which takes into account, school closures, workplace closures, travel bans etc.. and gives a value from 0 to 100, with 100 being a very high stringency index. Sri Lanka had a stringency index over 75% in 11/21 months from April 2020 to December 2021 (Fig 3A). The dengue case numbers seen from March 2020 to December 2021 were inversely proportional to the stringency index of Sri Lanka (Spearman's correlation coefficient r = -0.3755 p = 0.0587), although this was not significant (Fig 3B). It was observed that the dengue cases showed a corresponding rise in case numbers whenever the stringency measures were relaxed (Fig 3A).

From March 2020 to October 2021 Sri Lankan schools were fully closed for a total of 49 weeks and partially closed (only some areas had schools open, or only children of certain grades attended school) for 22 weeks, amounting to a total of 71 weeks of some form of school closure. From March 2020 to August 2020, Sri Lankan schools were fully closed for 21 weeks. From September 2020 to August 2021, Sri Lankan schools were fully closed for 20 weeks and partially closed for 17 weeks and from September 2021 to October 2021 schools were partially closed for 5 weeks. After the number of COVID-19 cases started to decline in October 2021, the schools partially reopened, the dengue cases that were already on the rise, further increased towards the end of the year (Fig 4A). The school closures showed a significant negative correlation with a change in dengue cases in 2021 (Spearman correlation coefficient; r = -0.4732, p = <0.0001) (Fig 4B).

## Relationship between dengue case number with mosquito indices

To determine whether the reduction in dengue cases during 2020 and 2021 was due to a reduction in mosquito breeding sites, we determined the association between the dengue cases with container and premise indices during this period. There was a positive but insignificant correlation between the dengue case numbers and *Aedes aegypti* premise index (Spearman corrélation coefficient; r = 0.8827, p = 0.93) and *Aedes aegypti* container index (Spearman correlation coefficient; r = 0.3825, p = 0.4667) (Fig 5A and 5B). There was no correlation seen between dengue cases and *Aedes albopictus* premise index (Spearman correlation coefficient; r = 0.2, p = 0.7139) and *Aedes albopictus* container index (Spearman correlation coefficient; r = -0.2571, p = 0.6583) (Fig 5C and 5D).

## Discussion

In this study we have explored the relationship between the changes in DENV serotypes, stringency index, school closures and the vector indices with the number of reported cases of dengue in the Colombo district, Sri Lanka from 2020 to 2021. We found that while there was no relationship between the vector indices and the DENV serotypes with the number of dengue cases, the numbers inversely correlated with the stringency index (non-significant trend) and the school closures.

Towards the end of 2019, due to the emergence of DENV3 in Sri Lanka after 10 to 15 years, there was a significant rise in the number of dengue cases, and this increase was seen until end of January 2020 [14]. By the end of 2019, DENV3 accounted for 28.9% of infections in

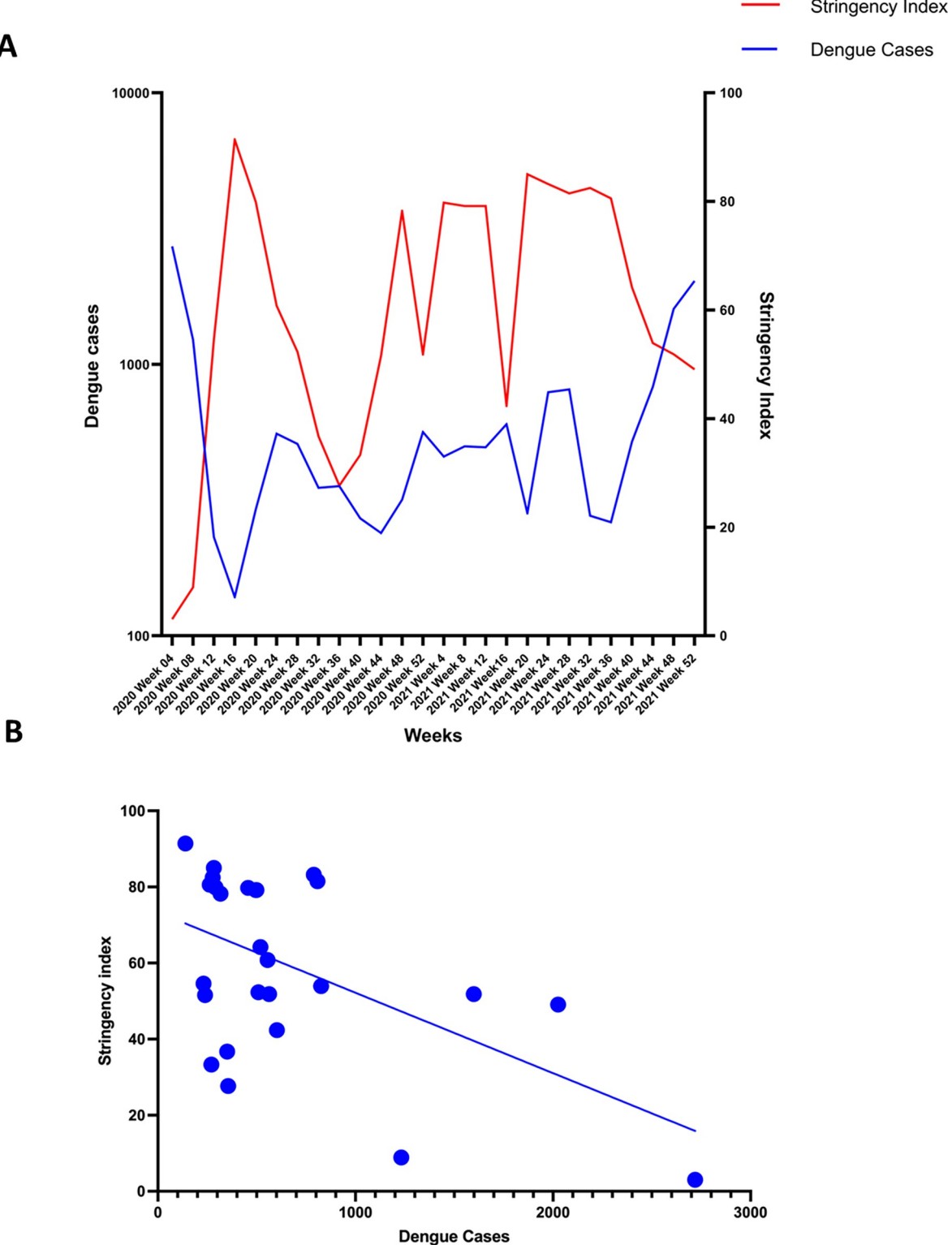

**Fig 3.** The relationship between the stringency index and the number of dengue cases reported from 2020 to 2021(A) and the correlation between the stringency index and number of dengue cases from 2020–2021 (B). The country stringency index during the years 2020 and 2021 was obtained from Our World in Data (6) and plotted against the number of dengue cases reported each month from all regions in Sri Lanka. The Spearman's correlation coefficient was measured (Spearman's r = -0.3755 p = 0.0587).

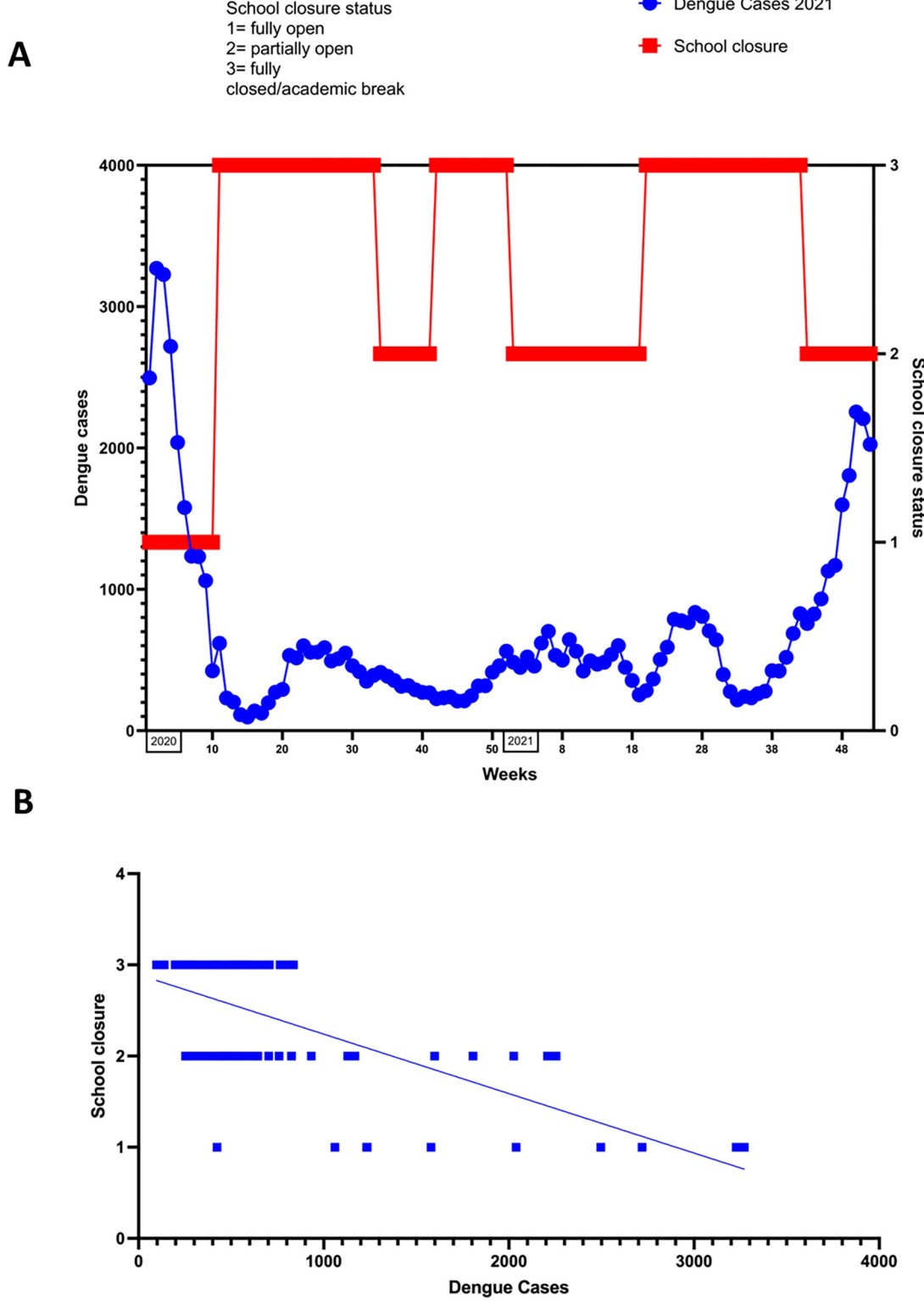

**Fig 4.** The relationship between number of dengue cases and periods of full and partial school closure in 2021(A) and the correlation between number of dengue cases and periods of full and partial school closure (B). The number of cases of dengue for 2021 were obtained from DenSys (2) and the SARS-CoV2 case numbers were obtained from Health Promotion Bureau (5). The data was plotted against school closure data for 2021 from UNESCO (7). School closure status was classified as 1 = fully open, 2 = partially open and 3 = fully closed/academic break. Spearman correlation was calculated (Spearman's r = - 0.4732, p = <0.0001).

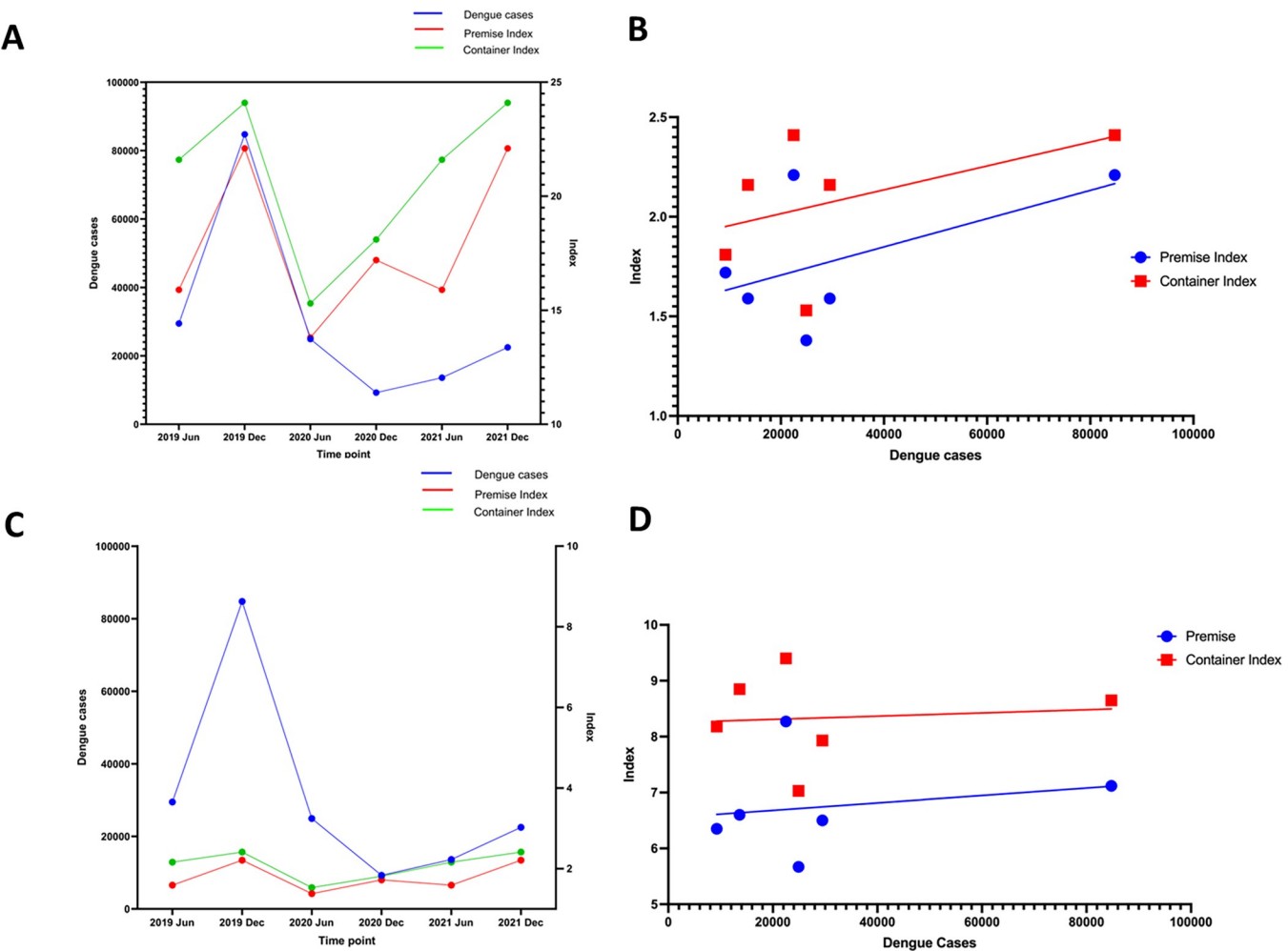

**Fig 5. The relationship between the number of dengue cases with mosquito indices.** Changes in the number of dengue cases with the *Aedes aegypti* and *Aedes albopictus* and container and premise indices data obtained from NDCU vector surveillance from January 2019 to December 2021 was plotted. The container and premise indices Aedes aegypti (A) and the correlation between the number of cases in vector indices *Aedes aegypti* (B) showed no correlation with the number of dengue cases with premise index (Spearman's r = 0.8827, p = 0.93) and container indices (Spearman's r = 0.3825, p = 0.4667). The relationship between the number of dengue cases with container and premise indices *Aedes albopictus* (C) and the correlation between the number of cases in vector indices *Aedes albopictus* (D) again showed no association seen with the number of dengue cases with premise index (Spearman's r = 0.2, p = 0.7139) or with the container index (Spearman's r = -0.2571, p = 0.6583).

Colombo, and when dengue cases started to increase by August 2021, the same frequency of the different DENV serotypes were seen. Therefore, despite a reduction in the number of cases in 2020 (79.4% less than in 2019) and in 2021 until August, there was no difference in the frequency of the circulating DENV serotypes. However, although 12.2% of the DENVs were of DENV1 serotype, the patients who had developed DHF were only found to be infected with either DENV2 or DENV3. Therefore, it is possible that certain types of DENV strains are associated with increased disease severity, as seen with previous outbreaks in different countries [15, 16].

Based on our data, it was the stringency index and most importantly the school closures that had the most significant impact on the dengue case numbers, as previously reported in another study [3]. Many schools in Sri Lanka and other tropical and sub-tropical countries are naturally ventilated and have classrooms open to the environment. This provides the ideal

environment for a day biting mosquito vector such as *Aedes aegpti* to obtain blood meals and therefore, infect as many children as possible [17]. Aedes mosquitoes have shown to engage in multiple feeding, which would be facilitated in a school environment due to the close proximity of hosts to one another, thereby reducing the distance the vector needs to fly to obtain another blood meal [18]. Although the importance of children gathering in schools in the transmission of dengue has not been previously studied, in one study in Mexico, it was shown that an intensive campaign in schools to eliminate mosquito breeding sites reduced the dengue incidence by 45% in the schools, while there was a 81% increase in the incidence in the country [19]. Since these data show that schools are probably one of the most important places of dengue transmission in the community, more focus should be given to vector control activities in schools to reduce the burden of dengue.

In the study done by Chen et al [3] in many Asia and Latin American countries and the study by Liyanage et al [6], the possible impact of changes in DENV serotype and the mosquito indices were not assessed. In our study, although we did not find any association with the mosquito indices and the number of dengue cases, a in 2020, a marked reduction in the mosquito indices is seen in the months of June, along with the reduction in dengue cases. This is surprising, given that in all previous years, the highest number of dengue cases are seen between the months of May to August, which coincides with the monsoons in the Western province [20]. Although the highest number of dengue cases have coincided with the seasonal monsoon, previous studies have also shown that there was no association seen with the number of dengue cases and the vector indices [20]. A limitation of this study is that we did not analyze the dengue cases by age to determine whether school closures directly affected a drop in the number of dengue infections among children. In addition, we did not have separate data on vector indices in schools and residences to determine any differences on the vector indices in schools compared to residences during periods where there was a high stringency index compared to a lower index. This information would help us to further understand effect of movement restriction on dengue epidemiology and vector indices.

In summary, our data show that the gatherings at school and human mobility are likely to be the main drivers of transmission of the DENV. However, emergence of new DENVs serotypes in a relatively non-immune population can also increase the dengue cases numbers as previously seen in many countries. Therefore, vector control strategies should focus mainly on vector control in school premises, while early warning systems should be established in countries to monitor the change in the circulating DENV serotypes.

## Supporting information

**S1 Data. Data used to generate the figures.**
(XLSX)

## Author Contributions

**Conceptualization:** Dinuka Ariyaratne, Graham S. Ogg, Gathsaurie Neelika Malavige.

**Data curation:** Dinuka Ariyaratne, Lahiru Kodituwakku, Anoja Dheerasinghe.

**Formal analysis:** Dinuka Ariyaratne, Gathsaurie Neelika Malavige.

**Investigation:** Laksiri Gomes, Tibutius T. P. Jayadas, Heshan Kuruppu.

**Project administration:** Nimalka Pannila Hetti, Anoja Dheerasinghe, Sudath Samaraweera.

**Resources:** Chandima Jeewandara, Nimalka Pannila Hetti, Sudath Samaraweera, Graham S. Ogg.

**Supervision:** Chandima Jeewandara, Gathsaurie Neelika Malavige.

**Writing – original draft:** Dinuka Ariyaratne, Gathsaurie Neelika Malavige.

**Writing – review & editing:** Graham S. Ogg, Gathsaurie Neelika Malavige.

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
