## [Decision Letter · Decision Letter 0]

12 May 2022

PGPH-D-22-00503

Epidemiological and virological factors determining dengue transmission in Sri Lanka during the COVID-19 pandemic

Dear Dr. Malavige,

Thank you for submitting your manuscript to PLOS Global Public Health. After careful consideration, we feel that it has merit but does not fully meet PLOS Global Public Health’s publication criteria as it currently stands. Therefore, we invite you to submit a revised version of the manuscript that addresses the points raised during the review process.

Please submit your revised manuscript by . If you will need more time than this to complete your revisions, please reply to this message or contact the journal office at globalpubhealth@plos.org. Please include the following items when submitting your revised manuscript:

We look forward to receiving your revised manuscript.

Kind regards,

Everton Falcão de Oliveira, Ph.D

Academic Editor

Journal Requirements:

1. You indicated that you had ethical approval for your study. In your Methods section, please ensure you have also stated whether you obtained consent from parents or guardians of the minors included in the study or whether the research ethics committee or IRB specifically waived the need for their consent.

State the initials, alongside each funding source, of each author to receive each grant.

3. Please ensure that the funders and grant numbers match between the Financial Disclosure field and the Funding Information tab in your submission form. Note that the funders must be provided in the same order in both places as well.

4. Please update your Competing Interests statement. If you have no competing interests to declare, please state: “The authors have declared that no competing interests exist.”

5. We do not publish any copyright or trademark symbols that usually accompany proprietary names, eg (R), (C), or TM  (e.g. next to drug or reagent names). Please remove all instances of trademark/copyright symbols throughout the text, including TaqMan® on page 7.

Reviewers' comments:

Reviewer's Responses to Questions

**Comments to the Author**

1. Does this manuscript meet PLOS Global Public Health’s publication criteria? Is the manuscript technically sound, and do the data support the conclusions? The manuscript must describe methodologically and ethically rigorous research with conclusions that are appropriately drawn based on the data presented.

Reviewer #1: Yes

Reviewer #2: Partly

2. Has the statistical analysis been performed appropriately and rigorously?

Reviewer #1: Yes

Reviewer #2: No

3. Have the authors made all data underlying the findings in their manuscript fully available (please refer to the Data Availability Statement at the start of the manuscript PDF file)?

Reviewer #1: Yes

Reviewer #2: Yes

4. Is the manuscript presented in an intelligible fashion and written in standard English?

Reviewer #1: Yes

Reviewer #2: Yes

5. Review Comments to the Author

Reviewer #1: Thank you for your important report. The article is well written, and the results do support the conclusion. This work differs from others, including serotype analyses and vector index analyses, demonstrating that social distancing measures may impact dengue transmission regardless of the virus serotype predominance and vector density.

Reviewer #2: This study aims to determine the association between the case numbers in Colombo with school closures, stringency index, changes in dengue virus (DENV) serotypes, and vector densities. Overall, this is an important study, but some points need to be improved and clarified.

The authors did not provide the line number on the manuscript. It would be very helpful for the reviewer if there were line numbers.

Page 2: In the abstract, you mentioned that this research aims to study the contribution of virological factors, human mobility, school closure, and mosquito factors in affecting the changes in dengue transmission in Sri Lanka. In the methods, you mentioned the stringency index. What is the relationship between the stringency index and human mobility? I would suggest defining stringency measures! What are the indicators of the stringency index?

Page 2 under Methods and findings: p=<0.0001. It should be p <=0.0001 or p � 0.001

In Introduction section, please provide more explanation regarding the important of this study and the novelty of this study.

Page 3: …mobility[6]. Please put a space!

Page 5: …DenSys[8]. Please put a space!

Page 6: … DenSys[8]. Please put a space!

Page 6: … Our World in Data[11]. Please put a space!

Page 6: … December 2021[12]. Please put a space!

Page 7 lines 1 -11: Please revise all the formulas!

Page 7: Please define CDC as it is first mentioned

Page 7 under the Real Time qPCR….: …DENV[13]. Please put a space!

Page 7: Please be consistent in writing DEN 1-4 or DENV, DENV1, DENV2, DENV3

Page 8: Under Statistical Analysis:

Please complete the statistical analysis so that it covers all statistical analyses used in this study. How did you analyze the variable of school closures and human mobility? It is not mentioned in the statistical analysis

Page 8: DEN 1, DEN2, DEN 3. Please be consistent!

Page 10: “Sri Lanka had a stringency index over 75% in 11/21 months from April 2020 to December 2021 (Figure 3A).” What does it mean?

Page 10, last sentence: …and partially closed for 22 weeks, amounting to a total of 71 weeks of some form of school closure. What do you mean by partially closed? Why did it happen?

Page 11: “The school closures significantly correlated with a change in dengue cases in 2021 (Spearman correlation coefficient; r=-0.4732, p=<0.0001”. What does it mean? How do you interpret it? Please explain more!

Page12 line 1: ..as possible[17]. Please put a space!

I would suggest including some limitations of this study and some potentials future works.

Figure 2A: what does the x-axis represent?

Please increase the resolution of Figure 3A!

How do you interpret Figure 4A? You define the “school closure status of 1, 2 and 3, but you did not define 4.

Please increase the resolution of Figure 5!

6. PLOS authors have the option to publish the peer review history of their article (what does this mean?). If published, this will include your full peer review and any attached files.

**Do you want your identity to be public for this peer review?** For information about this choice, including consent withdrawal, please see our Privacy Policy.

Reviewer #1: **Yes: **Gabriel Berg de Almeida

Reviewer #2: No

---

## [Decision Letter · Decision Letter 1]

21 Jun 2022

Epidemiological and virological factors determining dengue transmission in Sri Lanka during the COVID-19 pandemic

PGPH-D-22-00503R1

Dear Professor Malavige,

We are pleased to inform you that your manuscript 'Epidemiological and virological factors determining dengue transmission in Sri Lanka during the COVID-19 pandemic' has been provisionally accepted for publication in PLOS Global Public Health.

Best regards,

Everton Falcão de Oliveira, Ph.D

Academic Editor

Reviewer Comments (if any, and for reference):

Reviewer's Responses to Questions

**Comments to the Author**

1. If the authors have adequately addressed your comments raised in a previous round of review and you feel that this manuscript is now acceptable for publication, you may indicate that here to bypass the “Comments to the Author” section, enter your conflict of interest statement in the “Confidential to Editor” section, and submit your "Accept" recommendation.

Reviewer #1: All comments have been addressed

Reviewer #2: All comments have been addressed

2. Does this manuscript meet PLOS Global Public Health’s publication criteria? Is the manuscript technically sound, and do the data support the conclusions? The manuscript must describe methodologically and ethically rigorous research with conclusions that are appropriately drawn based on the data presented.

Reviewer #1: Yes

Reviewer #2: Yes

3. Has the statistical analysis been performed appropriately and rigorously?

Reviewer #1: Yes

Reviewer #2: Yes

4. Have the authors made all data underlying the findings in their manuscript fully available (please refer to the Data Availability Statement at the start of the manuscript PDF file)?

Reviewer #1: Yes

Reviewer #2: Yes

5. Is the manuscript presented in an intelligible fashion and written in standard English?

Reviewer #1: Yes

Reviewer #2: Yes

6. Review Comments to the Author

Reviewer #1: The authors have adressed all issues arised.

Reviewer #2: All comments from reviewer have been addressed

7. PLOS authors have the option to publish the peer review history of their article (what does this mean?). If published, this will include your full peer review and any attached files.

**Do you want your identity to be public for this peer review?** For information about this choice, including consent withdrawal, please see our Privacy Policy.

Reviewer #1: No

Reviewer #2: No
